# Neural implicit mapping via nested neighborhoods: real-time rendering of textured neural SDFs

## Abstract

We introduce a framework to address the problem of real-time joint estimation of surface **geometry** and its **attributes** (normals and textures) from neural SDFs. This problem was only partially approached by previous works, which do not support attributes nor dynamic surfaces in real-time. The framework is built on the **nesting condition**, which establishes a criteria for the neighborhoods of zero-level sets of a small sequence of neural SDFs to be nested. This allows mappings between such neighborhoods, enabling the definition of algorithms to use multiple neural SDFs to increase sphere tracing performance, while being able to compute the surface normals efficiently, and to map attributes between the surfaces. This framework **does not use spatial data-structures** and its components, besides real-time rendering of dynamic neural SDFs, can be used to **augment meshes with smooth neural normals and textures**. Our GEMM-based normal computation **does not depend on auto-differentiation nor computational graphs**, resulting in a performance improvement.

## 1 Introduction

*Neural signed distance functions* (SDFs) are an emerging model representation in Computer Graphics. They are coord-based neural networks approximating SDFs of surfaces, which can be rendered by finding their zero-level sets. *Sphere tracing* (ST) (Hart et al., 1989; Hart, 1996) is a standard algorithm for that task. It depends on evaluating the SDF many times along each view ray, which may be prohibitive in a real-time rendering context if the SDF is neural. Additionally, after estimating the hit points, it is necessary to compute their normals for the light simulation. Auto-differentiation is employed when using neural SDFs, a process that requires a computational graph, which may be unavailable in a CUDA renderer, and imposes additional overhead. Finally, a complete 3D rendering pipeline should consider textures. However, the current literature is still lacking in works that approach this component for level sets of neural SDFs in a real-time context.

An alternative to render neural SDFs is to extract a triangle mesh using marching cubes (Lorensen & Cline, 1987). However, due to the non-real-time nature of marching cubes, its application may be prohibitive in certain contexts involving neural SDFs. For instance, rendering dynamic SDFs requires mesh extraction at each time instant of the animation, posing a memory challenge if performed as preprocessing (extracted meshes use hundreds of MB in high resolutions while neural SDFs use a few KB). The performance is also a concern when surfaces are extracted while the animation is playing. Additionally, the smooth nature of the level sets of neural SDFs, which is a desired property in rendering, may be compromised by discretizing them into triangle meshes. ST offers a solution to capture this smoothness. The algorithm is employed for rendering complex shapes, such as fractals (Hart et al., 1989) and may use recent ray tracing hardware (da Silva et al., 2021). Advancing neural SDF rendering through ST is an important topic rooted in classic Computer Graphics.

In this work, we approach the problem of real-time rendering of level sets of dynamic neural SDFs including attributes (detailed normals and textures) using ST. That problem is crucial for Computer Graphics because level sets of neural SDFs are becoming a common surface representation. Thus, rendering them in real-time with support for dynamic behavior and attributes is an important task.

We propose a framework to address the aforementioned problem. It is based on *nested neighborhoods* and mappings between them. We define three algorithms: multiscale ST, neural attribute mapping, and a GEMM-based computation of normals. The *multiscale ST* uses the neighborhoods to minimize the overhead of iterations, using coarse versions of the SDF for acceleration. We provide experiments showing that it allows rendering of level sets of neural SDFs in real-time. Additionally, our framework supports dynamic SDFs, which would otherwise need level set extractions for each animation frame. We also dismiss the need for spatial data structures.

To represent attribute functions on level sets of SDFs we define the *neural attribute mapping*. It maps attributes from a neural SDF to another, usually a coarser one. This procedure is smooth and easy to apply since it does not depend on parameterizations. As a result, it can also fetch attributes for discrete triangle meshes or skip later ST iterations. The neural attribute mapping is well-defined because we require the attribute function to be constant along the streamlines of the gradient of the SDF, restricted to a (tubular) neighborhood. This mapping can also be used in addition to marching cubes, as the neural attributes may be applied directly at the mesh vertices without any parameterization.

Finally, we propose a *GEMM-based analytical computation of normals*, which does not need auto-differentiation nor a computational graph. We compute the surface normals in a modified forward pass of the network without the dependence of a machine learning library to do so. That means the algorithm may be used on a common renderer with access to a GEMM library.

## 2 RELATED WORK

Implicit functions are an essential topic in computer graphics (Velho et al., 2007). SDFs are important examples of such functions (Bloomenthal & Wyvill, 1990) and arise from solving the Eikonal problem (Sethian & Vladimirsky, 2000). Recently, neural networks have been used to model SDFs (Park et al., 2019; Gropp et al., 2020). *Sinusoidal networks* (SIRENs) Sitzmann et al. (2020) are an example of such, being *multilayer perceptrons* (MLPs) using sine as their activation function. In this work, we adopt the framework for training SIRENs proposed in (Novello et al., 2022).

*Marching cubes* (Lorensen & Cline, 1987; Lewiner et al., 2003) and ST (Hart et al., 1989; Hart, 1996) are classical visualization methods for rendering level sets of SDFs. Neural versions of those algorithms were proposed by (Liao et al., 2018; Chen & Zhang, 2021; Liu et al., 2020). While the initial works in neural SDFs use marching cubes to generate visualizations of the resulting level sets (Gropp et al., 2020; Sitzmann et al., 2020; Park et al., 2019), recent performance-driven approaches have been using ST, since no intermediary representation is needed for rendering (Davies et al., 2020; Takikawa et al., 2021). Our proposed multiscale ST considers a similar path.

**Surface representations and rendering:** Recent works propose (neural) surface representations disentangling base geometry and details. Wang et al. (2022) describe a representation using base and displacement networks to compute a final detailed surface, which are extracted via marching cubes for rendering. Morreale et al. (2022) employ coord-based neural networks to model surfaces parametrically. In this case, the networks are used to represent the surface parameterizations. Differently from our approach, those methods do not deal with the rendering problem, but establish surface representations. Sharp & Jacobson (2022) describe a way to perform geometric queries for neural implicit surfaces using range analysis. One of those queries is ray casting to render surfaces. However, the approach is not real-time (evaluations are in the order of seconds). None of those approaches support dynamic surfaces nor textures.

**Real-time neural SDFs:** Fast inference is needed to sphere trace neural SDFs in real-time. Davies et al. (2020) show that this is possible using General Matrix Multiply (GEMM) (Dongarra et al., 1990; Müller, 2021), but the capacity of their networks does not represent geometric detail. Other works in neural SDFs store features in the nodes of *octrees* (Takikawa et al., 2021; Martel et al., 2021), or limit the frequency band in training as in BACON (Lindell et al., 2021). However, octree-based approaches cannot directly handle dynamic models. NGLOD (Takikawa et al., 2021) is the reference real-time approach for rendering neural SDFs. It uses a *sparse voxel octree* (SVO) to represent the neural SDF and render its zero-level set using a *sparse* ST algorithm. Specifically, the vertices of the voxels store features. Then for a point $p$ and a level $L$ of the SVO, the features are trilinearly interpolated inside each voxel containing $p$ up to the level $L$. The resulting $L$ interpolated points are summed and passed as input to a neural network $f_{\theta_L}$. Thus, besides the SVO structure, NGLOD uses

a sequence of $L$ networks to represent the levels of detail. Moreover, the interpolation may result in neural SDFs with non-continuous gradients at the voxels boundaries leading to discretization artifacts (see Sec. 5.1). Additionally, NGLOD does not address real-time rendering of dynamic surfaces nor textures as our method does. Our approach supports continuous normals, by leveraging sinusoidal networks to fit each level of the SDFs using (Novello et al., 2022). Additionally, we use the extension in (Novello et al., 2023) to train dynamic SDFs.

Our approach is flexible regarding the surface representation which is uncoupled from the rendering. Given a neural SDF parameterized by BACON, or a sequence of SIRENs representing level-of-detail, our method renders them without additional training (Sec. 3.3). On the other hand, rendering using NGLOD would require the computation of the SVO and training the whole sequence of MLPs $f_{\theta_L}$.

**Attribute mapping:** *Normal mapping* (Cohen et al., 1998; Cignoni et al., 1998) is a classic method to transfer detailed normals between meshes, inspired by *bump mapping* (Blinn, 1978) and *displacement mapping* (Krishnamurthy & Levoy, 1996). Besides depending on interpolation, normal mapping also suffers distortions of the parameterization between meshes, which are assumed to have the same topology. Using the smooth properties of neural SDFs allows us to map the gradient of a finer neural SDF to a coarser one. This mapping considers a volumetric neighborhood of the coarse level set instead of parameterizations not relying on interpolations like the classic one. Inspired by the classical approach in (Bertalmío et al., 2001), we define the attribute function on a neighborhood of a the zero-level set of a surface by considering it constant along the normals.

*Texture mapping* (Catmull, 1974) is a technique for cost-effective rendering that involves mapping images onto surfaces using parameterizations. In the domain of neural rendering, Texture Fields (Oechsle et al., 2019) shares similarities with our neural attribute mapping but differs in input and color spaces. Our method processes a mesh with UV-mapped textures, while Texture Fields demand a 3D shape and an object image, using view dependent depth map rendering. Our approach uses a surface's tubular neighborhood, to define color along normals and is independent of viewpoints. Texture Fields, though effective, is non-real-time due to its use of 4 or 6 ResNet blocks and complex networks for latent code generation, surpassing the complexity of Texture Fields itself. In contrast, our method adopts simpler architectures like MLPs for efficient representation. GET3D (Gao et al., 2022) uses Texture Fields for the textures in its 3D model generation, sharing an analogous contextualization.

# 3 NESTED NEIGHBORHOODS OF NEURAL SDFS

Given the iterative nature of ST, a way to increase its performance is to optimize or avoid iterations. We propose to use neural SDFs with a small number of parameters to approximate earlier iterations and mapping the normals and the texture of the desired neural SDF, avoiding later iterations. Both tasks can be accomplished by mapping neural SDFs using nested neighborhoods.

## 3.1 OVERVIEW

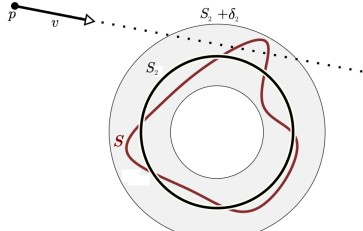

Figure 1: Overview. *Multiscale ST*: to sphere trace $S_2$ we first sphere trace the boundary of a neighborhood of $S_1$ (gray), resulting in $q_1$. Then we continue to sphere trace $S_2$, reaching $q_2$. *Neural attribute mapping*: since $q_2$ belongs to a (tubular) neighborhood of $S$, we evaluate the normal $N$ at $q_2$ of a parallel surface of $S$ (red dotted). These surfaces share the same normals. Texture color is acquired by making it constant along the line $q + tN$.

Figure 2: Illustration of a ray intersecting a surface $S$ nested in a neighborhood of another surface.

The basic idea comes from the following fact: if the zero-level set of a neural SDF $f$ is contained in a neighborhood $V$ of the zero-level set of another neural SDF, then we can map $f$ into $V$. We use an example with three SDFs as an overview and follow the notation in Fig. 1. Let $S_1, S_2, S$ be surfaces pairwise close with SDFs $h_1, h_2, f$ sorted by complexity. We use $S_1$ and $S_2$ to illustrate the multiscale ST and $S$ to illustrate the attribute mapping.

**Multiscale ST:** Suppose that the ray $p + tv$, with origin at a point $p$ and direction $v$, intersects $S_2$ at a point $q_2$. To compute $q_2$, we first sphere trace the boundary of a neighborhood of $S_1$ (gray) containing $S_2$, by using $f_1$. This results in $q_1$. Then we continue to sphere trace $S_2$ using $h_2$, reaching $q_2$. In other words, we are mapping the values of $h_2$ to the neighborhood of $S_1$. Using induction allows us to extend this idea to a sequence of SDFs (Sec. 3.2).

**Neural Attribute Mapping:** For shading, we need a normal at $q_2$, obtained by evaluating $N_2 = \nabla h_2(q_2)$. Instead, we propose to pull the finer details of $S$ to $S_2$ to increase fidelity. This is done by mapping the normals from $S$ to $S_2$ using $N = \nabla f(q_2)$. To justify this choice, note that $q_2$ belongs to a neighborhood of $S$. Thus, $N$ is the normal of $S$ at its closest point $q = q_2 - \epsilon N$, where $\epsilon$ is the distance $f(q_2)$ from $q_2$ to $S$. By doing so, we transfer the normal $N$ of $S$ at $q$ to $q_2$. Observe that $N$ is also the normal of the $\epsilon$-level set of $f$ at $q_2$ (red dotted). Similarly, the texture color is mapped from $q$ to $q_2$ by making it constant along $q + tN$ between $q$ and $q_2$.

## 3.2 DEFINITION

A *neural SDF* $f : \mathbb{R}^3 \to \mathbb{R}$ is a smooth neural network such that $|\nabla f| \approx 1$. We call its *zero-level set* a *neural surface* and denote it by $S$. In this work, we deal with the problem of rendering $S$ using the sphere tracing algorithm. Thus, given a point $p_0$ and a direction $v$, we must iterate $p_{i+1} = p_i + v f(p_i)$. However, evaluating $f(p_i)$ may be prohibitive for real-time applications, thus we use a coarse neural SDF (with less complexity) $h : \mathbb{R}^3 \to \mathbb{R}$ for the early iterations of the algorithm; $h \approx f$.

For the above proposal to work, we need $S$ to be *nested* in a $\delta$-neighborhood of $h^{-1}(0)$, i.e. $S \subset [|h| \leq \delta]$. Thus, we ray trace $h^{-1}(\delta)$ iterating $p_{i+1} = p_i + v h(p_i)$ and continue the iterations in the $\delta$-neighborhood using the target SDF $f$ (see Fig 2). Therefore, if the ray $p_0 + tv$ intersects $S$, the above procedure converges.

To use more neural SDFs we need an additional condition. Specifically, let $h_1$ and $h_2$ be coarser neural SDFs (sorted by complexity) approximating $f$. To extend the above procedure to work on the sequence $h_1, h_2, f$, we can first sphere trace a coarser level set $h_1^{-1}(\delta_1)$, then, $h_2^{-1}(\delta_2)$, and finally, $S$. For such algorithm to converge, we need $[|h_2| \leq \delta_2]$ to be *nested* in $[|h_1| \leq \delta_1]$, otherwise, we may miss the hit point (See Fig. 3 (b)). The choice of $\delta_1$ and $\delta_2$ values plays an important role on rendering. Having different values for them is also necessary to avoid issues (see Fig. 3).

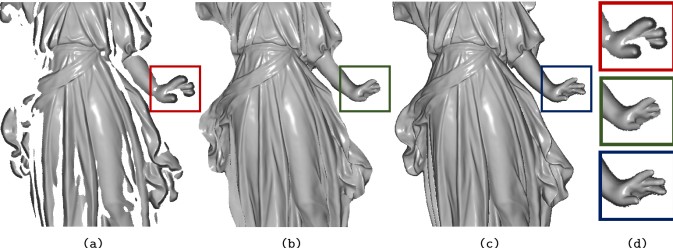

Figure 3: Practical $\delta_i$ choice. The number of ST iterations is fixed. (a) $\delta_i$ are equal to $0.02$, resulting in holes. More iterations would be needed using the finer (more complex) SDF to fill those holes, defeating the idea to minimize iterations. (b) $\delta_i$ are equal to $0.001$. Now the holes are closed, but parts of the silhouette are missing (notice the hand). (c) $\delta_1 = 0.02$ and $\delta_2 = 0.006$. In this case the nesting condition is satisfied, implying in no holes and a better silhouette capture.

In practice, we may choose how to use functions $h_2$ and $f$ to adapt to a specific performance budget. For example, we may choose not to iterate using $h_2$, but directly map the normals of the surface in $f$ to the surface in $h_1$. This setup relies on sphere tracing iterations using the coarse SDF, decreasing the cost per iteration. Another possible case is to use $h_2$ for a better surface approximation, and map normals of the surface of $f$ as before. We can also use $h_2$ and $f$ for sphere tracing and do not perform normal mapping. This setup has the best silhouette. Section 5.2 presents an evaluation of those cases.

We can extend these sequences to support evolutions of neural SDFs. For this, suppose that the sequence of networks $h_1, h_2, f$, has the *space-time* $\mathbb{R}^3 \times \mathbb{R}$ as domain. Then, we require the sequence of neural SDFs $h_1(\cdot, t), h_2(\cdot, t), f(\cdot, t)$ to satisfy the above nesting condition, for each $t$. Varying $t$ animates the sequence. See Appx A.2 for more details on training dynamic SDFs.

### 3.3 NESTING THE NEIGHBORHOODS

This section describes approaches to create sequences of neural SDFs with nested neighborhoods. The objective is to train a sequence of neural SDFs sorted by inference time and to find small upper-bound thresholds that ensure the nesting condition.

**BACON** (Lindell et al., 2021) is a multiresolution network to represent neural SDFs. Its LODs are sorted by inference time and can be used to define our neural SDFs. Specifically, let $h_1$ and $h_2$ be LODs of a BACON $f$, and $\epsilon > 0$ be a small number. Defining $\varepsilon_1 = |h_1 - h_2|_\infty + \epsilon$ and $\varepsilon_2 = |h_2 - f|_\infty + \epsilon$ results in $|h_1 - h_2|_\infty < \varepsilon_1$ and $|h_2 - f|_\infty < \varepsilon_2$. Then, Prop 1 gives the thresholds $\delta_i$ implying that $h_1, h_2$ and $f$ satisfy the nesting condition.

**Prop. 1.** *Let $h$ and $f$ be neural SDFs satisfying $|h - f|_\infty < \varepsilon$ for $\varepsilon > 0$. Thus, for each $\delta > 0$ we have that $\big[|h| < \delta\big] \subset \big[|f| < \varepsilon + \delta\big]$.*

Prop 1 is a consequence of the following fact: for each $p \in \big[|h| < \delta\big]$, we have

$$|f|(p) = |f(p) - h(p) + h(p)| \leq \underbrace{|f(p) - h(p)|}_{<\varepsilon} + \underbrace{|h(p)|}_{<\delta} < \varepsilon + \delta.$$

**MLPs for a single surface:** By the *spectral bias* (Rahaman et al., 2019), MLPs tend to learn lower frequencies first. Thus, we propose training MLPs $h_1$ and $h_2$ with increasing capacity and frequency range (SIREN's $\omega_0$) to approximate the finer neural SDF $f$, resulting in a sequence of neural SDFs sorted by inference time and by detail representation capacity. As in the BACON case, we can define thresholds $\delta_i$ using Prop 1 resulting in a sequence of nested neural SDFs $h_1, h_2$ and $f$.

To compute $\delta_j$ we need to evaluate the infinity norm on the training domain of the neural network. In practice, we approximate it using the maximum absolute difference in a sample $p_i$ (Sec. 5.2).

We train $h_1, h_2, f$ from fine to coarse and use the previously trained SDFs as ground truth for the next. That is, we consider $f$ as ground truth for $h_2$, which in turn is ground truth for $h_1$. During the training of $h_2$, we evaluate $|h_2 - f|_\infty$, and choose the training step with the smallest norm. We do the same for $h_1$. We note that the resulting norm $|h_2 - f|_\infty$ is small during most of the training, a consequence of $|h_2 - f|_2$ being a term in our loss function. Additionally, the above procedure provides a nested sequence that gives a robust rendering. Besides the resulting deltas being small (0.02 and 0.006 in Fig. 3), we note that decreasing them results in loss of silhouette (Fig. 3 (c)).

### 3.4 MULTISCALE SPHERE TRACING

Let $f$ be a neural SDF, and $\{h_i\}$ be a sequence of $m \leq 2$ auxiliary neural SDFs such that $\{h_i\}, f$ is a nested sequence. Let $p$ be a point and $v$ be a direction, we define the multiscale ST (Alg. 1) to approximate the intersection (if it exists) between $S$ and $\gamma(t) = p + tv$, with $t > 0$.

Specifically, we assume $p \notin \big[|h_1| \leq \delta_1\big]$. The multiscale ST is based on the fact that to sphere trace $S$ we can first sphere trace $h_m^{-1}(\delta_m)$ using $h_m$ (Fig. 2). Lines 3-6 describe the ST of $h_j^{-1}(\delta_j)$ for $j = 1, \ldots, m$ (line 1). If $j = m$ we sphere trace $S$ instead of its neighborhood (line 4). In the dynamic case, the algorithm operates on time-dependent nested neural SDFs.

---

**ALGORITHM 1:** Multiscale ST

**Input:** Sequence of nested neural SDFs $\{h_i\}$, $f$, point $p$, direction $v$, threshold $\epsilon > 0$
**Output:** End point $p$

```
1  for j = 1, ..., m do
2      t = +∞;
3      while t > ε do
4          t = (j==m)?f(p):h_j(p) − δ_j ;
5          p = p + tv;
6      end
7  end
```

If $\gamma \cap S \neq \emptyset$, the ST approximates the first hit point between $\gamma$ and $S$. This is due to the nesting condition, which ensures that if $\gamma \cap S \neq \emptyset$ implies $\gamma \cap h_2^{-1}(\delta_2) \neq \emptyset$, and then $\gamma \cap h_1^{-1}(\delta_1) \neq \emptyset$.

For the inference of a neural SDF, in Line 4 of Alg. 1, we use the GEMM alg. (Dongarra et al., 1990) for each layer. To finish the rendering, we need to compute the normals and the textures.

## 4 NEURAL ATTRIBUTE MAPPING

Let $S$ be a surface nested in a $\delta$-neighborhood of the zero-level set of a neural SDF $h$, that is, $S \subset \big[|h| \leq \delta\big]$. Let $g : \big[|h| \leq \delta\big] \to \mathcal{C}$ be an attribute function represented by a neural network. The *neural attribute mapping* assigns to each $p \in S$ the attribute $g(p)$. If $\nabla h$ has no critical points in $\big[|h| \leq \delta\big]$, we can connect $p$ to a point $q \in h^{-1}(0)$ by integrating $-\nabla h$. We assume that the function $g$ is constant along the resulting path. This property is characterized by $\langle \nabla g, \nabla h \rangle = 0$ which we use

as a constraint during the training of $g$. Therefore, such a procedure maps the attributes defined on $h^{-1}(0)$ to the surface $S$. Next, we present two applications: normal mapping and texture mapping.

## 4.1 NEURAL NORMAL MAPPING

We introduce *neural normal mapping* as an example of attribute mapping. Assume $f$ to be a finer neural SDF having $S$ nested on its $\delta$-neighborhood. The *neural normal mapping* assigns to each $p \in S$ the attribute $g(p) := \nabla f(p)$. This is a restriction of $\nabla f$ to $S$ and maps the normal of $f^{-1}(0)$, along the minimum path connecting it to $p$. The attribute $g$ is constant along the path since $\|\nabla f\| = 1$ (Fig. 1).

We explore two cases. First, let $S$ be a triangle mesh. We use the neural normal mapping to transfer the detailed normals of the level sets of $f$ to the vertices of $S$. This approach is analogous to the classic normal mapping, which maps detailed normals stored in textures to meshes via parameterizations. However, practical UV parameterization demands highly-manual UV unwrapping. Since our method is volumetric and automatic such parameterizations are not needed (see Fig. 7 - middle).

For the second case, let $S$ be the zero-level set of another coarse neural SDF. We can use the neural normal mapping to avoid the overhead of additional ST iterations (see Fig. 7 - left). In this case, we do not need to extract a surface using marching cubes, which may generate meshes with prohibitive resolutions for real-time applications and cannot extract animations from dynamic neural SDFs. Animated neural SDFs are supported by mapping the normals of $f(\cdot, t)$ onto the animated surface.

## 4.2 NEURAL TEXTURE MAPPING

We follow the above notation and consider $S$ to be a surface nested in the $\delta$-neighborhood of a neural SDF $f$. We define an attribute function $g : \mathbb{R}^3 \to \mathcal{C}$ to encode a *texture* on the $\delta$-neighborhood of $f$. In this case, we consider the codomain $\mathcal{C}$ to be the RGB space. We denote the attribute mapping associated to the triple $\{S, f, g\}$ a *neural texture mapping*. Such mapping is responsible for transferring the texture defined on the $\delta$-neighborhood to the surface $S$. To train the set of parameters $\phi$ of the network $g$ we use the following loss functional: $\mathcal{L}(\phi) = \int_{f^{-1}(0)} (g - \mathfrak{g})^2 dx + \int_{\left[ |f| \leq \delta \right]} \langle \nabla g, \nabla f \rangle^2 dx$. where the first term forces $g$ to fit to the *ground-truth* texture $\mathfrak{g}$, and the second term asks for $g$ to be constant along the gradient paths, that is, it regularizes the network on the $\delta$-neighborhood of $f$.

## 4.3 GEMM-BASED ANALYTICAL NORMAL CALCULATION FOR MLPS

We propose a GEMM-based analytical computation of normals, which are continuous and do not need auto-differentiation. This results in smooth normals, as shown in Fig 4.

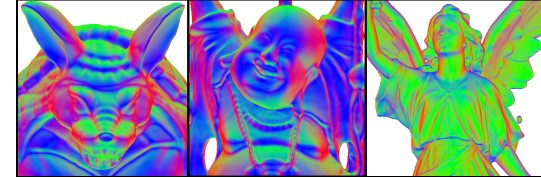

Our derivation develops directly from the *chain rule* applied to the MLP. Recall that a MLP with $n - 1$ hidden layers has the following form:

$$f(p) = W_n \circ f_{n-1} \circ \cdots \circ f_0(p) + b_n, \quad (1)$$

Figure 4: Rendering of the normals calculated using our approach. They are naturally smooth as a consequence of working on the continuous setting.

where $f_i(p_i) = \varphi(W_i p_i + b_i)$ is the $i$-layer. The *activation* $\varphi$ is applied on each coordinate of the linear map $W_i : \mathbb{R}^{N_i} \to \mathbb{R}^{N_{i+1}}$ translated by $b_i \in \mathbb{R}^{N_{i+1}}$. The gradient of $f$ is given using the *chain rule*:

$$\nabla f(p) = W_n \cdot \mathbf{J} f_{n-1}(p_{n-1}) \cdot \cdots \cdot \mathbf{J} f_0(p), \quad \text{with} \quad \mathbf{J} f_i(p_i) = W_i \odot \varphi' \left[ a_i | \cdots | a_i \right] \quad (2)$$

$\mathbf{J}$ is the *Jacobian*, $p_i := f_{i-1} \circ \cdots \circ f_0(p)$, $\odot$ is the *Hadamard* product, and $a_i = W_i(p_i) + b_i$. Eq. 2 is used in (Gropp et al., 2020; Novello et al., 2022) to compute the level set normals analytically.

We now use Eq. 2 to derive a GEMM-based algorithm for computing the normals of $S_\theta$ in real-time. Those normals are given by $\nabla f$ which is a sequence of matrix multiplications. These multiplications are not appropriate for a GEMM setting because $\mathbf{J} f_0(p) \in \mathbb{R}^{3 \times N_1}$. The GEMM algorithm organizes the input points into a matrix, where its lines correspond to the points and its columns organize them and enable parallelism. We can solve this problem using three GEMMs, one for each normal coordinate. Thus, each GEMM starts with a column of $\mathbf{J} f_0(p)$, eliminating one of the dimensions. The resulting multiplications can be asynchronous since they are completely independent.

The $j$-coord of $\nabla f$ is given by $G_n = W_n \cdot G_{n-1}$, where $G_{n-1}$ is given by iterating $G_i = \mathbf{J} f_i(p_i) \cdot G_{i-1}$, with the initial condition $G_0 = W_0[j] \odot \varphi'(a_0)$. The vector $W_0[j]$ denotes the $j$-column of $W_0$. We use a kernel and a GEMM to compute $G_0$ and $G_n$. For $G_i$ with $0 < i < n$, observe that

$$G_i = (W_i \odot \varphi' [a_i | \cdots | a_i]) \cdot G_{i-1} = (W_i \cdot G_{i-1}) \odot \varphi'(a_i).$$

The first equality comes from Eq. 2 and the second from a commutative property of the Hadamard product. The second expression needs fewer computations and is solved using a GEMM followed by a kernel. Please refer to Appx A.1 for a detailed algorithm.

## 5 EXPERIMENTS

We compare each part of our framework against SOTA methods and present ablation studies. For the sphere tracing related experiments, we fix the number of iterations for better control of parallelism. All experiments are conducted on an NVidia Geforce RTX 3090.

We use a simplified notation to refer to the MLPs. Here, $(64, 1) \triangleright (256, 3)$ means a neural SDF sequence with a MLP with one $64 \times 64$ matrix (2 hidden layers with 64 neurons), and a MLP with three $256 \times 256$ matrices (4 hidden layers with 256 neurons). Another example: $(64, 1)$ is a single MLP.

| Model | Nets | Dist. |
|---|---|---|
| Arm. | (64,1) | 0.0035 |
| | (256,3) | 0.0021 |
| Bunny | (64,1) | 0.0024 |
| | (256,1) | 0.0019 |
| | (256,3) | 0.0021 |
| Buddha | (64,1) | 0.0051 |
| | (256,1) | 0.0019 |
| | (256,3) | 0.0016 |
| Lucy | (64,1) | 0.0071 |
| | (256,1) | 0.0024 |
| | (256,3) | 0.0017 |

Table 1: Hausdorff dist. between the trained models and the ground-truth.

### 5.1 COMPARISONS

**Surface:** Our first set of experiments compare our neighborhood-nesting approach against SOTA methods for surface representation and rendering, namely implicit displacement fields (IDF) (Wang et al., 2022), and NGLOD (Takikawa et al., 2021). IDF has SOTA quality for surface representations that disentangle shape and detail. NGLOD is the reference for real-time rendering of neural SDFs.

| Neural Armadillo | Training (s) |
|---|---|
| (64, 1) | 23.6 |
| (256, 2) | 95.1 |
| (256, 3) | 128.7 |
| (64, 1) ▷ (256, 2) | **118.7** |
| (64, 1) ▷ (256, 3) | 152.3 |
| IDF | **100.1** |
| NGLOD | 1628.0 |

Table 2: Although our method is real time for rendering, its training time is comparable to IDF. IDF depends on marching cubes to render, losing the smooth properties of neural SDFs. Our training is also faster than NGLOD.

Table 2 compares the training times. Even though our rendering is real-time, we have comparable training times against IDF, which needs to extract a parametric surface before rendering. Our training is one order of magnitude faster than NGLOD.

Figure 5 shows a rendering comparison. 5a uses the real-time configuration for NGLOD, recommended by the authors in the repository documentation. As discussed in Sec. 2, its formulation results in non-continuous normals, causing discretization artifacts. We also present a non-real-time LOD 5 configuration (5b), which has less discretization artifacts. 5c shows our real-time neural normal mapping applied into a coarse surface. Since our approach works on the smooth setting, we support smooth normals. Fig 5d shows the surface generated by IDF, after a surface extraction using marching cubes with 512 resolution. Note that IDF and NGLOD do not support dynamic surfaces nor textures. IDF would need to train a surface and extract it for each time step to be dynamic. Analogously, NGLOD would need to create octrees and train surface extractors for each time step. Fig 8 shows that our approach runs in real time for dynamic surfaces.

**Normals:** We compare our GEMM-based normal calculation against PyTorch's autograd. As shown in Table 4 (Appx A.1), our technique performs around $2\times$ faster. We tested 6 different networks trained for Armadillo, Happy Buddha, and Lucy, varying between 2 and 3 hidden layers.

**Textures:** Since our approach is the first to address textures for neural SDFs in real-time, we compare against the ground-truth meshes with UV-textures. Table 7 shows the MSE between the images generated by our method and the ground-truth. Please refer to Fig 10 (Appx A.3) for the images used to compute the MSEs. Fig 6 shows the neural texture mapping applied to coarse surfaces.

Since our method defines the textures in a neighborhood of the surface, no parameterization or UV map is needed. Inference is simple and consists of a single MLP evaluation for a batch of points. The results show that our approach achieves good appearance while uncoupling it from geometry in a compositional manner.

### 5.2 ABLATION STUDIES

Figure 5: Render comparison.

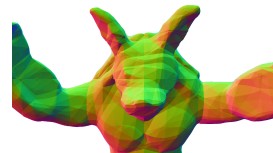

(a) NGLOD LOD 0 (real-time). Note the discretization artifacts (mosaic appearance).

**Infinity norm and surface quality:** We evaluate the infinity norm approximation (Sec. 3.3). We performed experiments to estimate it by varying the sample sizes (1K, . . ., 200K) used in the approximation. We sampled these points in a $\Delta$-neighborhood of the zero-level set. We use $\Delta = 0.1, \cdots, 0.5$ which provides appropriate neighborhoods, since our training domain is restricted to $[-1, 1]^3$. For each measurement, we sample a batch of points uniformly. The reported distances are the averages of the distances computed over 1000 evaluations of the above tests for each case. Table 5 (in Appx A.3) shows that the computation is robust and does not require a large sample.

To evaluate the efficiency of coarse neural SDFs to represent the ground-truth SDFs, Table 1 shows the Hausdorff distances between their neural level sets and the original meshes. All distances are within the third decimal digit, which means they are very close to the ground-truth. This fact corroborates our assumption that coarse surfaces in nested neighborhoods can be used to accelerate their rendering.

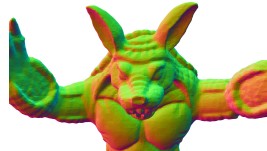

(b) NGLOD LOD 5 (non-real-time). Less artifacts.

**Neural normal mapping and multiscale ST:** Regarding image quality/perception, Fig. 7 shows the case where the coarse surface is the zero-level of a neural SDF (left) and when it is a triangle mesh (middle). An overall evaluation of the algorithm with other models is given in Fig. 9 (Appx A.3). In all cases, normal mapping increases fidelity.

The quantitative results corroborate that statement. Table 6 in Appx A.3 shows time/memory/MSE measured in a Python render. Note that the normal mapping increases fidelity (up to $30\%$ MSE improvement in comparison with the coarse Armadillo) and speed up the rendering (up to 6x improvement in comparison with the baseline).

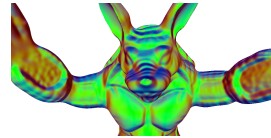

(c) Ours (real-time). Normal mapping of a fine $(256, 3)$ SDF into a coarse $(64, 1)$ SDF. Note the smooth normals.

The result may be improved using the multiscale ST, as shown in Fig 7 (right). Adding ST iterations using a neural SDF with a better approximation of the surface improves the silhouette. This is aligned with the results presented in Table 6. The last two rows of each example show cases with iterations in the second SDF, up to 20x improvement in comparison with the pure normal mapping for Lucy.

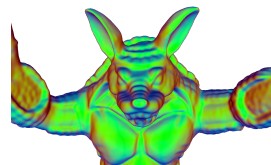

(d) IDF (not real-time). Surface extracted using MC.

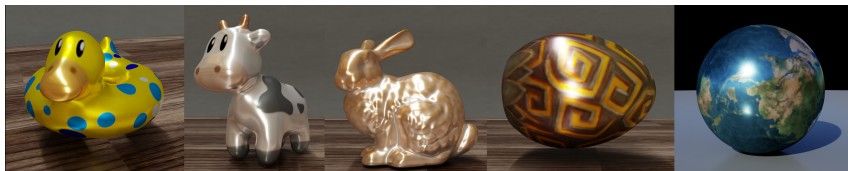

Figure 6: Neural texture mapping. All networks are $(256, 3)$, except for the the earth, which is $(512, 3)$. The surfaces are marching cubes of $(64, 1)$ SDFs, except for the bunny, which is $(128, 2)$. No parameterization or UV map is needed.

**Real-time renderer:** We evaluate a GPU version implemented in a CUDA renderer, using neural normal mapping, multiscale ST, and the GEMM-based analytical normal calculation (implemented using CUTLASS). Table 3 shows the results. Notice that the framework achieves real-time performance and that using neural normal mapping and multiscale ST improves performance considerably.

**Dynamic SDFs:** Fig. 8 shows an evaluation of a dynamic SDF interpolating the Falcon and Witch. The baseline neural SDF is $(128, 2)$ and the coarse is $(64, 1)$. The normal mapping case runs at 73 FPS using CUDA. See the supp. video for the animation and additional examples.

## 6 CONCLUSION

We presented a novel approach for real-time joint estimation of surface geometry and its attributes (normals and textures). It supports dynamic SDFs and does not need spatial data structures. The

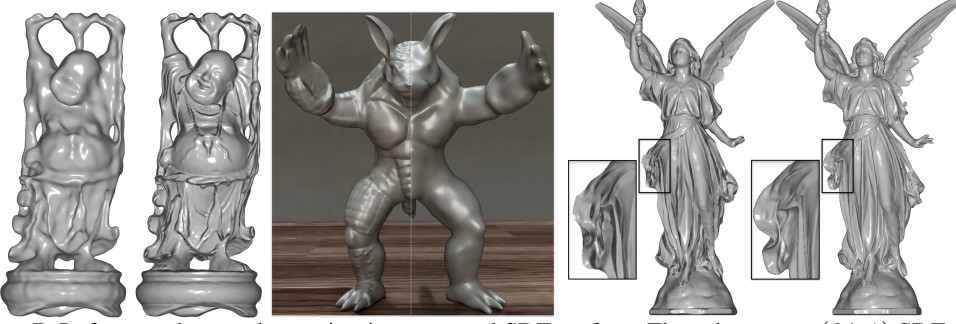

Figure 7: Left: neural normal mapping into a neural SDF surface. First, the coarse $(64, 1)$ SDF. Then, the neural normal mapping of the $(256, 3)$ SDF into the $(64, 1)$. Middle: neural normal mapping into half of a triangle mesh. The normals of the $(256, 3)$ SDF are used. The mesh is the marching cubes of the $(64, 1)$ SDF. The *mean square error* (MSE) is 0.00262 for the coarse case and 0.00087 for the normal mapping, an improvement of 3x. The baseline is the marching cubes of the $(256, 3)$ SDF. Right: Silhouette evaluation. First a $(64, 1) \triangleright (256, 3)$, then a $(64, 1) \triangleright (256, 2) \triangleright (256, 3)$ configuration. Notice how the silhouette improves with the additional $(256, 2)$ level.

multiscale ST accelerates the surface evaluation, the neural attribute mapping transfers surface attributes from a neural SDF to another surface, and the GEMM-based analytical normal computation provides smooth normals without the need of auto-differentiation. Neural attribute mapping has potential to impact 3D content generation workflows, since it does not need any parameterization such as UV mapping.

This approach opens paths for future works. One possibility is exploring other attribute mappings, such as BRDFs, and hypertextures. Any attribute that lies in a neighborhood of a surface is a potential candidate. Multiscale ST could also probably be applied into neural SDF-based 3D reconstruction or inverse rendering tasks to reduce the training time. Nested neighborhoods could probably be adapted for unsigned distance functions too. Improvements can be done for further performance optimization. For example,

| Model | FPS | Speedup | Mem |
|---|---|---|---|
| $(256, 3)$ | 19.8 | 1.0X | 777 |
| $(64, 1)$ | **128.8** | **6.5X** | 18 |
| $(64, 1) \triangleright (256, 1)$ | **73.1** | **3.7X** | 281 |
| $(64, 1) \triangleright (256, 2)$ | **53.0** | **2.7X** | 538 |
| $(64, 1) \triangleright (256, 3)$ | **41.6** | **2.1X** | 795 |
| $(64, 1) \triangleright (256, 1) \triangleright (256, 3)$ | **39.1** | **2.0X** | 1058 |

Table 3: Real-time SIREN evaluations using our GEMM normals in a CUDA renderer. The number of iterations is 20 for the first neural SDF and 5 for the second in the last row. The last network is used for normal mapping. Images are $512 \times 512$. Memory is in KB. All cases using multiscale ST and neural normal mapping result in speedups.

using fully fused GEMMs may decrease the overhead of GEMM setup (Müller, 2021). Approaches to minimize parameters in the networks could also impact the performance.

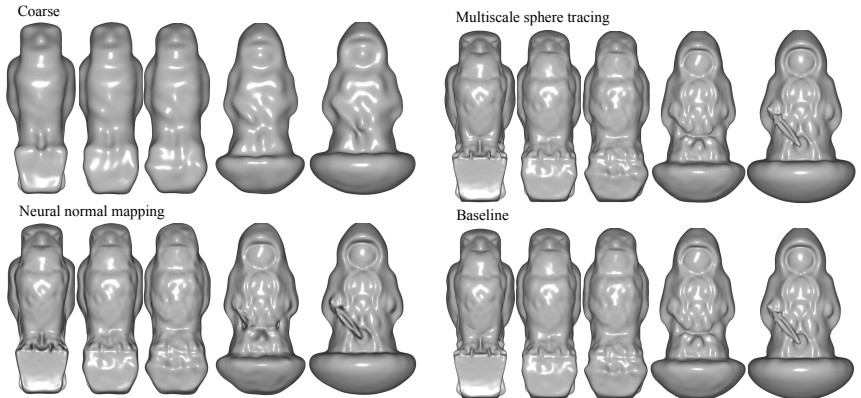

Figure 8: Interpolating Falcon and Witch. Coarse: $(64, 1)$, fine: $(64, 1) \triangleright (128, 2)$, and baseline: $(128, 2)$. Note that the normal mapping and the multiscale ST increase fidelity. Running at 73 FPS.

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

## A APPENDIX

### A.1 GEMM-BASED NORMAL COMPUTATION ALGORITHM

Algorithm 2 presents the gradient computation for a batch of points as described in Section 4.3. The input is a matrix $P \in \mathbb{R}^{3 \times k}$ with columns storing the $k$ points generated by the GEMM version of Algorithm 1. The algorithm outputs a matrix $\nabla f_\theta(P) \in \mathbb{R}^{3 \times k}$, where its $j$-column is the gradient of $f_\theta$ evaluated at $P[j]$. Lines $2 - 5$ are responsible for computing $G_0$, Lines $6 - 11$ compute $G_{n-1}$, and Line 13 provides the result gradient $G_n$. Table 4 shows a comparison between this algorithm and automatic differentiation using pytorch.

---

**ALGORITHM 2:** Normal computation

**Input:** neural SDF $f_\theta$, positions $P$
**Output:** Gradients $\nabla f_\theta(P)$

1 **for** $j = 0$ *to* 2 *(async)* **do**
2     using a GEMM: `// Input Layer`
3       $A_0 = W_0 \cdot P + b_0$
4     using a kernel:
5       $G_0 = W_0[j] \odot \varphi'(A_0); \; P_0 = \varphi(A_0)$
    `// Hidden layers`
6     **for** *layer* $i = 1$ *to* $n - 1$ **do**
7       using GEMMs:
8        $A_i = W_i \cdot P_{i-1} + b_i;$
       $G_i = W_i \cdot G_{i-1}$
9       using a kernel:
10        $G_i = G_i \odot \varphi'(A_i); \; P_i = \varphi(A_i)$
11     **end**
12     using a GEMM: `// Output layer`
13       $G_n = W_n \cdot G_{n-1}$
14 **end**

---

### A.2 NEURAL SDF TRAINING

For the sake of self-containment, we describe how we train the neural networks used in our experiments. The idea is to make an overview of the established approaches we use.

| Model | Autograd | Ours | Resolution |
|-------|----------|------|------------|
| Armadillo 256x2 | 0.007 | 0.003 | 512x512 |
| Armadillo 256x2 | 0.024 | 0.010 | 1024x1024 |
| Armadillo 256x3 | 0.010 | 0.005 | 512x512 |
| Armadillo 256x3 | 0.025 | 0.012 | 1024x1024 |
| Buddha 256x2 | 0.008 | 0.005 | 512x512 |
| Buddha 256x2 | 0.021 | 0.014 | 1024x1024 |
| Buddha 256x3 | 0.011 | 0.005 | 512x512 |
| Buddha 256x3 | 0.024 | 0.012 | 1024x1024 |
| Lucy 256x2 | 0.007 | 0.004 | 512x512 |
| Lucy 256x2 | 0.021 | 0.012 | 1024x1024 |
| Lucy 256x3 | 0.011 | 0.007 | 512x512 |
| Lucy 256x3 | 0.025 | 0.015 | 1024x1024 |

Table 4: Runtime comparison, in seconds, between Pytorch autograd and our algorithm to calculate the normals.

**Static surfaces:** We apply the method described in (Novello et al., 2022) to train a neural SDF. Specifically, let $S$ be a surface, and $f_\theta : \mathbb{R}^3 \to \mathbb{R}$ be a sinusoidal MLP. To compute the parameters $\theta$ such that $f_\theta^{-1}(0) \approx S$, it is common to consider the *Eikonal* problem:

$$1 - \|\nabla f_\theta\| = 0 \text{ subject to } f_\theta = 0 \text{ on } S. \quad (3)$$

Which asks for $f_\theta$ to be the SDF of a set containing $S$. It can be derived from Eq 3 that $\langle \nabla f_\theta, N \rangle = 1$ on $S$, which implies that $\nabla f_\theta$ must be aligned with the normals of $S$. Then, they use Eq (3) to define a loss function to train $f_\theta$.

$$\mathcal{L}(\theta) = \int_{\mathbb{R}^3} \left(1 - \|\nabla f_\theta\|\right)^2 dp + \int_S f_\theta^2 + \left(1 - \langle \nabla f_\theta, N \rangle\right) dS.$$

Where the first term encourages $f_\theta$ to be the SDF of a set $\mathcal{X}$, the second term encourages $\mathcal{X}$ to contain $S$ and asks for the alignment between $\nabla f_\theta$ and the normal field of $S$.

**Dynamic surfaces**

We consider the approach in (Novello et al., 2023) to evolve the SDF $g : \mathbb{R}^3 \to \mathbb{R}$ of the surface $S$. For this, the authors considered the *level set equation* (LSE) (Sethian, 1999) to model the implicit evolution of $g$. Thus, the domain of the neural SDFs must be extended to $\mathbb{R}^3 \times \mathbb{R}$, where the parameter $t \in \mathbb{R}$ controls the evolution. Specifically, they train $f_\theta : \mathbb{R}^3 \times \mathbb{R} \to \mathbb{R}$ by forcing it to approximate a

| samples | Delta | dist | samples | Delta | dist |
|---|---|---|---|---|---|
| 1000 | 0.1 | 0.0075 | 5000 | 0.1 | 0.0077 |
| 1000 | 0.2 | 0.0107 | 5000 | 0.2 | 0.0107 |
| 1000 | 0.3 | 0.0113 | 5000 | 0.3 | 0.0117 |
| 1000 | 0.4 | 0.0117 | 5000 | 0.4 | 0.0118 |
| 1000 | 0.5 | 0.0125 | 5000 | 0.5 | 0.0118 |
| 10000 | 0.1 | 0.0076 | 50000 | 0.1 | 0.0073 |
| 10000 | 0.2 | 0.0110 | 50000 | 0.2 | 0.0103 |
| 10000 | 0.3 | 0.0113 | 50000 | 0.3 | 0.0118 |
| 10000 | 0.4 | 0.0127 | 50000 | 0.4 | 0.0117 |
| 10000 | 0.5 | 0.0121 | 50000 | 0.5 | 0.0118 |
| 100000 | 0.1 | 0.0073 | 200000 | 0.1 | 0.0076 |
| 100000 | 0.2 | 0.0107 | 200000 | 0.2 | 0.0109 |
| 100000 | 0.3 | 0.0118 | 200000 | 0.3 | 0.0126 |
| 100000 | 0.4 | 0.0129 | 200000 | 0.4 | 0.0121 |
| 100000 | 0.5 | 0.0121 | 200000 | 0.5 | 0.0125 |

Table 5: Evaluation of the sup norm approximation. We vary the sample of points, in a Delta-neighborhood of the coarse zero-level set, for norm estimations. We also vary the Delta parameter to verify the robustness computation regarding points far away from the zero-level set.

solution of the LSE which is given as follows:

$$
\begin{cases}
\dfrac{\partial f_\theta}{\partial t} + v \, \|\nabla f_\theta\| = 0 & \text{in } \mathbb{R}^3 \times (a,b), \\
f_\theta = g & \text{on } \mathbb{R}^3 \times \{0\}.
\end{cases}
\tag{4}
$$

This equation evolves the level sets of $g$ towards their normals times a function $v$. The interval $(a,b)$ can be used to control the resulting neural animation $S_t$ of $S$. As in the static case, the LSE defines a loss function to train $f_\theta$.

$$
\mathcal{L}(\theta) = \underbrace{\int_{\mathbb{R}^3 \times (a,b)} \left( \frac{\partial f_\theta}{\partial t} + v \, \|\nabla f_\theta\| \right)^2 dp\,dt}_{\mathcal{L}_{\text{LSE}}(\theta)} + \underbrace{\int_{\mathbb{R}^3 \times \{0\}} \left( f_\theta - g \right)^2 dp}_{\mathcal{L}_{\text{data}}(\theta)}.
\tag{5}
$$

The constraint $\mathcal{L}_{\text{LSE}}$ forces $f_\theta$ to satisfy the LSE and works as a regularization of $f_\theta$ that requires it to follow the underlying deformation. The constraint $\mathcal{L}_{\text{data}}$ asks for $f_\theta$ to satisfies $f_\theta = g$ on $\mathbb{R}^3 \times \{0\}$.

### A.3 ADDITIONAL EXPERIMENTS

**Surface representation ablation:** Table 6 shows an extensive evaluation of the multiscale ST and neural normal mapping algorithms. We highlight the neural normal mapping and multiscale ST cases, which considerably improve the time and MSE performance, respectively.

**Infinity norm computation:** Table 5 shows the numbers used for the discussion in Section 5.1 about the computation of the infinity norm. The table shows that the computation is robust to variations in the size of the neighborhood and sample size.

**Broader perceptual evaluation:** On the paper we exemplify results using one model for each experiment. Fig. 9 shows a broader perceptual evaluation of the multiscale sphere tracing and the neural normal mapping using several models. Fig. 10 also shows the images we use to calculate the MSE to compare the neural texture mapping with the rendering baseline.

**Image resolution ablation test:** We perform quantitative evaluation tests, varying the image resolution. This is shown by Tables 8 through 10. All of them have the same structure: the first section is the coarse MLP, the second is the baseline MLP, the third is the multiscale sphere tracing (MS), and the fourth is the normal mapping (NM). The second column shows the number of sphere tracing iterations for each MLP, separated by commas in case the finer MLP is also iterated. We highlight the time in the normal mapping section and the mean square error (MSE) in the multiscale sphere tracing section to emphasize how those techniques impact the evaluation metrics.

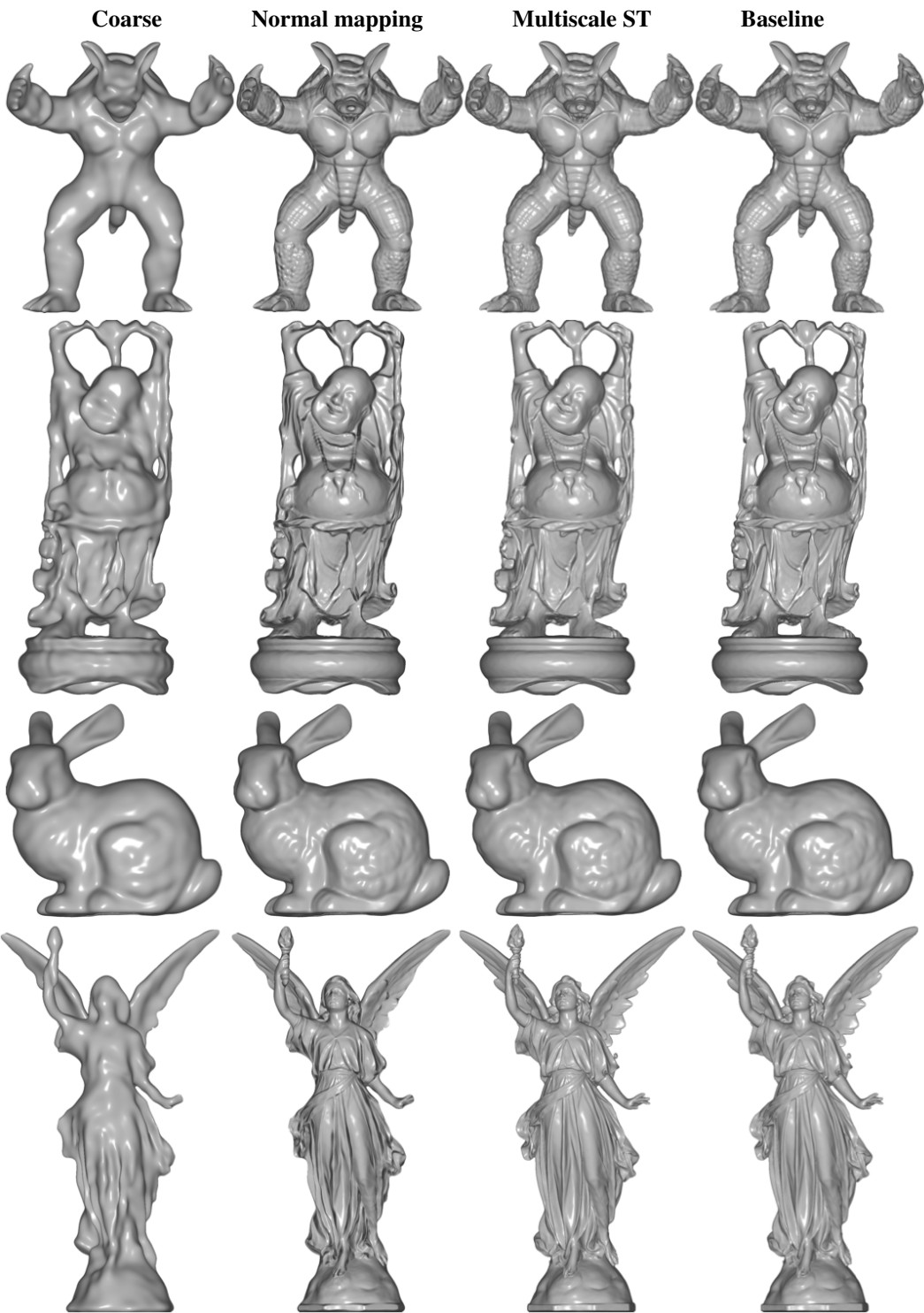

Figure 9: Comparison between our method and the SIREN baseline. The columns represent different configurations. From left to right: $(64, 1)$, $(64, 1) \triangleright (256, 1)$ (Bunny and Dragon) and $(64, 1) \triangleright (256, 2)$ (Happy Buddha and Lucy), $(64, 1) \triangleright (256, 3)$, and the baseline $(256, 3)$. The second column uses neural normal mapping and the third uses multiscale sphere tracing. Notice that fidelity is improved in the second column and the third column refines the results.

**Baseline**                    **Ours**

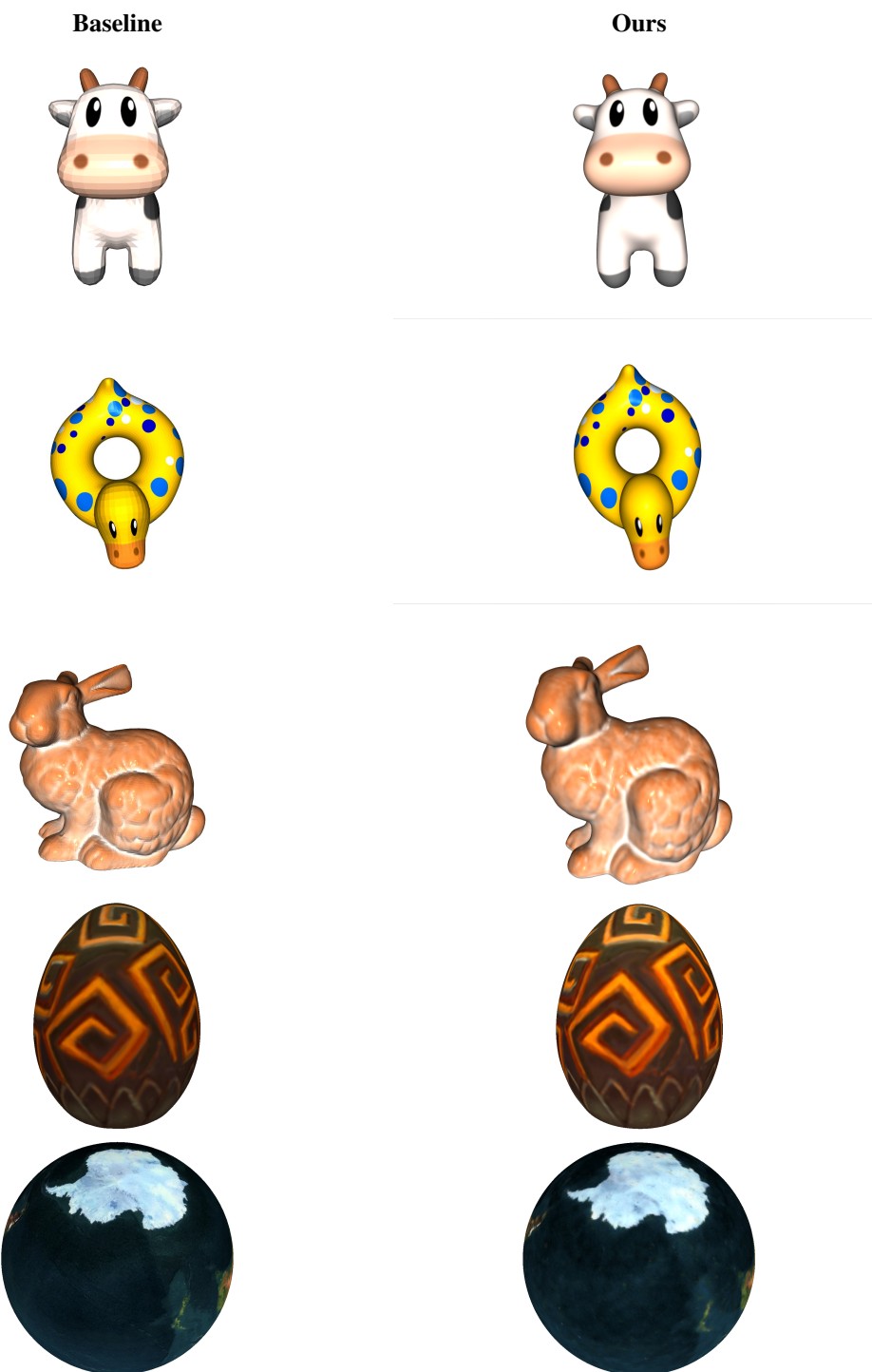

Figure 10: Images we use to calculate the MSE between the ground-truth textured meshes and our approach.

**SIRENs (left)**

| | SIRENs | Iters | Time | Mem | MSE |
|---|---|---|---|---|---|
| **Armadillo** | *(256,3)* | *40* | *2.442* | *777* | *-* |
| | (64, 1) | 40 | 0.298 | 18 | 0.00588 |
| | (64, 1) ▷ (256, 3) | 40, 0 | **0.409** | 795 | 0.00452 |
| | (256, 3) | 15 | 0.936 | 777 | 0.01237 |
| | (64, 1) ▷ (256, 3) | 30,10 | 0.895 | 795 | 0.00746 |
| | (64, 1) ▷ (256, 3) | 30,30 | 1.934 | 795 | **0.00057** |
| **Buddha** | *(256,3)* | *40* | *2.228* | *777* | *-* |
| | (64,1) | 40 | 0.299 | 18 | 0.00485 |
| | (64, 1) ▷ (256, 3) | 40, 0 | **0.413** | 795 | 0.00441 |
| | (256,3) | 15 | 0.928 | 777 | 0.00589 |
| | (64, 1) ▷ (256, 3) | 30,10 | 0.893 | 795 | 0.00355 |
| | (64, 1) ▷ (256, 3) | 30,30 | 1.945 | 795 | **0.00048** |
| **Bunny** | *(256,3)* | *40* | *2.237* | *777* | *-* |
| | (64,1) | 40 | 0.287 | 18 | 0.00229 |
| | (64, 1) ▷ (256, 3) | 40, 0 | **0.403** | 795 | 0.00191 |
| | (256,3) | 15 | 0.928 | 777 | 0.00793 |
| | (64, 1) ▷ (256, 3) | 30,10 | 0.886 | 795 | 0.00417 |
| | (64, 1) ▷ (256, 3) | 30,30 | 1.920 | 795 | **0.00065** |
| **Lucy** | *(256,3)* | *40* | *2.239* | *777* | *-* |
| | (64,1) | 40 | 0.312 | 18 | 0.00518 |
| | (64, 1) ▷ (256, 3) | 40, 0 | **0.420** | 795 | 0.00470 |
| | (256,3) | 15 | 0.941 | 777 | 0.00280 |
| | (64, 1) ▷ (256, 3) | 30,10 | 0.927 | 795 | 0.00363 |
| | (64, 1) ▷ (256, 3) | 30,30 | 1.977 | 795 | **0.00024** |

**BACON (right)**

| | BACONs | Iters | Time | MSE |
|---|---|---|---|---|
| **Armadillo** | *(256,6)* | *100* | *10.067* | *-* |
| | (256, 2) | 100 | 4.829 | 0.00473 |
| | (256, 2) ▷ (256, 6) | 100,0 | **4.945** | 0.00309 |
| | (256, 2) ▷ (256, 6) | 50,30 | 5.526 | 0.00061 |
| | (256, 2) ▷ (256, 6) | 50,50 | 7.438 | **0.00040** |
| **Buddha** | *(256,6)* | *100* | *9.851* | *-* |
| | (256, 2) | 100 | 4.836 | 0.00455 |
| | (256, 2) ▷ (256, 6) | 100,0 | **4.946** | 0.00284 |
| | (256, 2) ▷ (256, 6) | 50,30 | 5.520 | 0.00086 |
| | (256, 2) ▷ (256, 6) | 50,50 | 7.450 | **0.00077** |
| **Bunny** | *(256,6)* | *100* | *9.861* | *-* |
| | (256, 2) | 100 | 4.835 | 0.00458 |
| | (256, 2) ▷ (256, 6) | 100,0 | **4.952** | 0.00260 |
| | (256, 2) ▷ (256, 6) | 50,30 | 5.524 | 0.00025 |
| | (256, 2) ▷ (256, 6) | 50,50 | 7.455 | **0.00013** |
| **Lucy** | *(256,6)* | *100* | *9.871* | *-* |
| | (256, 2) | 100 | 4.852 | 0.00400 |
| | (256, 2) ▷ (256, 6) | 100,0 | **4.968** | 0.00207 |
| | (256, 2) ▷ (256, 6) | 50,30 | 5.559 | 0.00023 |
| | (256, 2) ▷ (256, 6) | 50,50 | 7.488 | **0.00018** |

Table 6: Evaluation of our method using two SIRENs (Sitzmann et al., 2020) (left) and BACON (Lindell et al., 2021) (right) in a Python render. Iters represent iterations used in each one (0 means no iteration and thus pure neural normal mapping). Time is in seconds and memory is in KB. The MSE is compared with the baseline (in italic). We emphasize in bold how the neural normal mapping has minimal time impact and how increasing iterations on the second SDF improves MSE. We use a BACON with 8 layers, with output in layers 2 and 6. For fairness, we use the same layers as SDFs for our method. All BACONs use 2151 KB of memory because the network contains all level of detail.

| Model | MSE |
|---|---|
| Spot | 0.0329 |
| Bob | 0.0434 |
| Bunny | 0.0720 |
| Egg | 0.0291 |
| Earth | 0.0033 |

Table 7: MSE between our textured images and the ground truth meshes.

| Armadillo (256x256) | Iters | Time(s) | MSE |
|---|---|---|---|
| (64, 1) | 15 | 0.0212 | 0.0184 |
| (64, 1) | 20 | 0.0238 | 0.0120 |
| (64, 1) | 30 | 0.0325 | 0.0067 |
| (64, 1) | 40 | 0.0426 | 0.0056 |
| (256, 3) | 15 | 0.0726 | 0.0168 |
| (256, 3) | 20 | 0.0894 | 0.0093 |
| (256, 3) | 30 | 0.1261 | 0.0023 |
| (64, 1) ▷ (256.3) MS | 30,10 | 0.0847 | 0.0102 |
| (64, 1) ▷ (256.3) MS | 30,20 | 0.1219 | 0.0035 |
| (64, 1) ▷ (256.3) MS | 30,30 | 0.1575 | **0.0007** |
| (64, 1) ▷ (256.3) NM | 40 | **0.0434** | 0.0046 |

| Armadillo (512x512) | Iters | Time(s) | MSE |
|---|---|---|---|
| (64, 1) | 15 | 0.0488 | 0.0183 |
| (64, 1) | 20 | 0.0586 | 0.0117 |
| (64, 1) | 30 | 0.0777 | 0.0067 |
| (64, 1) | 40 | 0.0973 | 0.0056 |
| (256, 3) | 15 | 0.2447 | 0.0166 |
| (256, 3) | 20 | 0.3307 | 0.0091 |
| (256, 3) | 30 | 0.4548 | 0.0026 |
| (64, 1) ▷ (256, 3) MS | 30,10 | 0.2706 | 0.0098 |
| (64, 1) ▷ (256, 3) MS | 30,20 | 0.3959 | 0.0037 |
| (64, 1) ▷ (256, 3) MS | 30,30 | 0.5183 | **0.0008** |
| (64, 1) ▷ (256, 3) NM | 40 | **0.1250** | 0.0047 |

| Armadillo (1024x1024) | Iters | Time(s) | MSE |
|---|---|---|---|
| (64, 1) | 15 | 0.1431 | 0.0182 |
| (64, 1) | 20 | 0.1713 | 0.0116 |
| (64, 1) | 30 | 0.2522 | 0.0067 |
| (64, 1) | 40 | 0.2891 | 0.0057 |
| (256, 3) | 15 | 0.9481 | 0.0165 |
| (256, 3) | 20 | 1.2108 | 0.0091 |
| (256, 3) | 30 | 1.7144 | 0.0026 |
| (64, 1) ▷ (256, 3) MS | 30,10 | 0.8893 | 0.0098 |
| (64, 1) ▷ (256, 3) MS | 30,20 | 1.4360 | 0.0037 |
| (64, 1) ▷ (256, 3) MS | 30,30 | 1.9357 | **0.0009** |
| (64, 1) ▷ (256, 3) NM | 40,40 | **0.4113** | 0.0048 |

Table 8: Image size evaluation for the Armadillo. We evaluate square images with resolution 256, 512, and 1024 pixels. The best time and MSE are highlighted. The MS suffix means multiscale ST, and the NM suffix means normal mapping.

| Buddha (256x256) | Iters | Time(s) | MSE |
|---|---|---|---|
| (64, 1) | 15 | 0.02371 | 0.01227 |
| (64, 1) | 20 | 0.02719 | 0.00911 |
| (64, 1) | 30 | 0.03248 | 0.00719 |
| (64, 1) | 40 | 0.04294 | 0.00713 |
| (256, 3) | 15 | 0.07295 | 0.00912 |
| (256, 3) | 20 | 0.09062 | 0.00473 |
| (256, 3) | 30 | 0.12203 | 0.00120 |
| (64, 1) ▷ (256, 3) MS | 30,10 | 0.08475 | 0.00550 |
| (64, 1) ▷ (256, 3) MS | 30,20 | 0.11944 | 0.00175 |
| (64, 1) ▷ (256, 3) MS | 30,30 | 0.16732 | **0.00049** |
| (64, 1) ▷ (256, 3) NM | 40 | **0.04733** | 0.00658 |

| Buddha (512x512) | Iters | Time(s) | MSE |
|---|---|---|---|
| (64, 1) | 15 | 0.05388 | 0.01221 |
| (64, 1) | 20 | 0.06792 | 0.00915 |
| (64, 1) | 30 | 0.08188 | 0.00734 |
| (64, 1) | 40 | 0.10191 | 0.00727 |
| (256, 3) | 15 | 0.24713 | 0.00895 |
| (256, 3) | 20 | 0.32203 | 0.00472 |
| (256, 3) | 30 | 0.44999 | 0.00117 |
| (64, 1) ▷ (256, 3) MS | 30,10 | 0.25617 | 0.00530 |
| (64, 1) ▷ (256, 3) MS | 30,20 | 0.39894 | 0.00177 |
| (64, 1) ▷ (256, 3) MS | 30,30 | 0.53451 | **0.00052** |
| (64, 1) ▷ (256, 3) NM | 40 | **0.12666** | 0.00668 |

| Buddha (1024x1024) | Iters | Time(s) | MSE |
|---|---|---|---|
| (64, 1) | 15 | 0.17758 | 0.01221 |
| (64, 1) | 20 | 0.18585 | 0.00911 |
| (64, 1) | 30 | 0.24527 | 0.00728 |
| (64, 1) | 40 | 0.30179 | 0.00718 |
| (256, 3) | 15 | 0.95683 | 0.00895 |
| (256, 3) | 20 | 1.20325 | 0.00474 |
| (256, 3) | 30 | 1.72849 | 0.00119 |
| (64, 1) ▷ (256, 3) MS | 30,10 | 0.87383 | 0.00535 |
| (64, 1) ▷ (256, 3) MS | 30,20 | 1.39137 | 0.00180 |
| (64, 1) ▷ (256, 3) MS | 30,30 | 1.95539 | **0.00054** |
| (64, 1) ▷ (256, 3) NM | 40 | **0.40734** | 0.00659 |

Table 9: Image size evaluation for Buddha. We evaluate square images with resolution 256, 512, and 1024 pixels. The best time and MSE are highlighted. The MS suffix means multiscale ST, and the NM suffix means normal mapping.

| Lucy (256x256) | Iters | Time(s) | MSE |
|---|---|---|---|
| (64, 1) | 15 | 0.02448 | 0.00673 |
| (64, 1) | 20 | 0.03082 | 0.00612 |
| (64, 1) | 30 | 0.03498 | 0.00615 |
| (64, 1) | 40 | 0.04696 | 0.00619 |
| (256, 3) | 15 | 0.08059 | 0.00397 |
| (256, 3) | 20 | 0.10305 | 0.00180 |
| (256, 3) | 30 | 0.13671 | 0.00032 |
| (64, 1) ▷ (256, 3) MS | 30,10 | 0.08307 | 0.00513 |
| (64, 1) ▷ (256, 3) MS | 30,20 | 0.12159 | 0.00117 |
| (64, 1) ▷ (256, 3) MS | 30,30 | 0.16263 | **0.00021** |
| (64, 1) ▷ (256, 3) NM | 40 | **0.04782** | 0.00602 |

| Lucy (512x512) | Iters | Time(s) | MSE |
|---|---|---|---|
| (64, 1) | 15 | 0.05538 | 0.00678 |
| (64, 1) | 20 | 0.07184 | 0.00625 |
| (64, 1) | 30 | 0.08435 | 0.00625 |
| (64, 1) | 40 | 0.10085 | 0.00637 |
| (256, 3) | 15 | 0.24135 | 0.00397 |
| (256, 3) | 20 | 0.32049 | 0.00189 |
| (256, 3) | 30 | 0.44454 | 0.00043 |
| (64, 1) ▷ (256, 3) MS | 30,10 | 0.24557 | 0.00526 |
| (64, 1) ▷ (256, 3) MS | 30,20 | 0.38773 | 0.00124 |
| (64, 1) ▷ (256, 3) MS | 30,30 | 0.53365 | **0.00026** |
| (64, 1) ▷ (256, 3) NM | 40 | **0.14132** | 0.00614 |

| Lucy (1024x1024) | Iters | Time(s)) | MSE |
|---|---|---|---|
| (64, 1) | 15 | 0.16215 | 0.00682 |
| (64, 1) | 20 | 0.19107 | 0.00628 |
| (64, 1) | 30 | 0.27376 | 0.00628 |
| (64, 1) | 40 | 0.31857 | 0.00642 |
| (256, 3) | 15 | 0.93262 | 0.00393 |
| (256, 3) | 20 | 1.23098 | 0.00185 |
| (256, 3) | 30 | 1.71587 | 0.00041 |
| (64, 1) ▷ (256, 3) | 30,10 | 0.96133 | 0.00520 |
| (64, 1) ▷ (256, 3) | 30,20 | 1.44421 | 0.00121 |
| (64, 1) ▷ (256, 3) | 30,30 | 1.95704 | **0.00025** |
| (64, 1) ▷ (256, 3) | 40 | **0.44357** | 0.00617 |

Table 10: Image size evaluation for Lucy. We evaluate square images with resolution 256, 512, and 1024 pixels. The best time and MSE are highlighted. The MS suffix means multiscale ST, and the NM suffix means normal mapping.

