# OpenReview forum: "Neural implicit mapping via nested neighborhoods: real-time rendering of neural SDFs with textures"
_ICLR.cc/2024/Conference — Submitted to ICLR 2024_

### Official Review · Reviewer_TKVb · 2023-10-18

**Soundness:** 2 fair
**Presentation:** 2 fair
**Contribution:** 2 fair
**Rating:** 3
**Confidence:** 5

**Summary:**

This paper proposes to use a neural implicit shape representation with multi-level of details, which allows fast sphere tracing: coarse, fast levels are traced first, followed by finer levels. Attributes, such as colors or normals can be transferred from one detailed level to a coarse level, for improved renderings in a constrained computational budget.

**Strengths:**

The idea of using multiple levels of details for efficiency is underexplored in the literature. Transferring details such as normals to a coarse surface is nice, and emulates the use of normal maps in the classical graphics methods.
Efficiency: The multiscale sphere tracing algorithm focuses on minimizing iteration time by using coarse approximations in earlier iterations. This can lead to faster rendering times and more efficient computations.
Analytic Normal Calculation: The paper proposes a fast algorithm for analytic normal calculation for MLPs, which can improve shading performance and accuracy during rendering.

**Weaknesses:**

The “neural attribute mapping” simply consists in evaluating a neural field on a given surface, for textures or normals. To be more formal, given a surface point s on surface S, and a neural field f:
- Case 1: if f encodes an RGB color, then s is colored with f(s).
- Case 2: if f encodes a surface, then s is given the normal ∇f(s). This case is interesting when S has a low level of details: by transferring the normals of the 0-levelset of f to S, one can render S with more details. But this is simply a nearest-neighbor assignment, no need for elaborate mathematics.

Both cases are very simple, and do not require the unnecessarily complex (pretentious?) formalism of “delta-nested neighborhood”, “integrating along the gradients”, “restriction of ∇f to S and mapping the normal along a path” presented in the paper.

Furthermore:
- Case 1 was already exploited in past papers, for example Texture Fields (Oechsle et al, ICCV 2019), GET3D ( Gao et al., NeurIPS 2022). It is very straightforward, and “achieving SOTA by uncoupling appearance from geometry in a compositional manner” is an overstatement. Additionally, the comparison to Instant NGP is irrelevant, as the problem setting is vastly different: Instant NGP learns a scene representation from images only. Here, the proposed method already has access to a supervision signal for the 3D geometry and for the RGB texture.
- Case 2 should be compared to a straightforward rendering of the more precise representation f, instead of transferring it. The speed benefits of a transfer of a high resolution field to a coarse surface should be exemplified. The only table reporting a speedup is Tab. 5, and it does not display a quality metric - only speedups.


The multi-scale sphere tracing algorithm is similar to the extension of marching cubes to multi-LOD presented in BACON. Its speed should be compared to extracting the surface (once) with marching cubes, and then rendering it with a rasterizer.

The GEMM implementation of normals computations should be compared to standard backpropagation in pytorch or tensorflow.
This is a component-wise computation of the normal vectors. The acronym GEMM is used several times without being defined, nor the concept. NeuS2 (Wang et al, ICCV 2023) also has a fast normal computation algorithm, it might be worth adding the reference.

Section 3.3 is supposed to “describe approaches to create sequences of neural SDFs with nested neighborhoods”. In fact, given any sequence of SDF networks, by choosing the epsilons large enough, we can nest them. The bounds derived in this section are very loose and can apply to any sequence, even for unrelated objects or surfaces. It would also work for levels of details in any order. In other words, saying that a sequence of neural surfaces is “nested” without characterizing the threshold for which it is nested has very little value: given any sequence, I can consider it as nested.

In the experiment section, when reporting training time vs NGLOD and IDF: which neural network architecture is used? Why call it “ours”? The whole method and contributions are about rendering neural fields with multiple levels of details, never about a new architecture or a faster training procedure. Why compare training times then?
Moreover, it is unclear if SIRENs or BACONs are used. A whole paragraph of the method section is dedicated to nesting BACONs, is it used here? The datasets should also be clarified in the main text.

Minor: when reporting MSE, scaling the values to improve readability and avoid all metrics starting with “0.00” would be better.

Clarity: the introduction is a bit of a circular definition: “the nested neighborhood model […] is a novel framework […] based on nested neighborhoods” is not explaining what “nested neighborhoods” are. Moreover, in many places the term "S is nested on the delta-neighborhood of S_theta" could simply be replaced by "S is in the delta-neighborhood of S_theta".



In conclusion, the idea of exploring multiple levels of details is Interesting, but its presentation uses overelaborate mathematics for very simple ideas that have already been exploited (learning a color field for texture for example).
Moreover, experimental results are not convincing. Comparisons with a traditional pipeline are lacking: since rendering speed is the main motivation, the approach should also thoroughly compared to a mesh rasterizer - and a one time execution of a CUDA implementation of marching cubes (like https://github.com/tatsy/torchmcubes).
While the method presents a formalism for “nesting” an arbitrary number of levels of details, most results are provided with a single network or 2 networks. Finally, clarity should be improved too (see questions and weaknesses).

**Questions:**

In which practical application should one consider using this multiscale SDF over a mesh rasterizer?

Could the definition of “nested surfaces” be simplified? For example, using a single threshold delta. I believe the following definition is equivalent to the one proposed:
“say that f_theta_2 is nested in f_theta_1 for threshold delta > 0 if S_theta_2 is contained in the delta-neighborhood of S_theta_1”

At the end of the first paragraph of Sec 5.1 : “the attribute g is constant along the path since ||∇f|| = 1”: why does a gradient with constant norm implies a constant gradient?

---

> ### Author Response · Authors · 2023-11-23
> **Part 1**
>
> We deeply thank the reviewer for the very valuable comments and insights. All questions were legitimate and we are happy because that resulted in considerable improvements in the paper. We politely ask the reviewer to read this new version. Next we discuss the changes based on the reviewer's comments.
>
> ### Formalism
> The reviewer was right about the text being too formal, compromising understanding. We changed our definitions in Section 3.2 to the minimum necessary. Now instead of a sequence with m neural SDFs, we have a sequence of 3 SDFs {h1, h2, f}, corresponding to the cases in our experiments. In practice, we may choose how to use functions h2 and f to adapt to a specific performance budget. For example, we may choose not to iterate using h2, but directly map the normals of the surface in f to the surface in h1. This setup relies on sphere tracing iterations using the coarse SDF, decreasing
> the cost per iteration. Another possible case is to use h2 for a better surface approximation, and map normals of the surface of f as before. We can also use h2 and f for sphere tracing and do not perform
> normal mapping. This setup has the best silhouette.
> Figure 3 was added to help explaining in practice why we need the definition of nested neighborhoods, why making the deltas equal is too restrictive, and whats happens when they are too big or too small. It shows that the rendering may end up with holes or the silhouette may be lost depending on the values.
>
> ### "Neural attribute mapping was already explored in other papers".
>
> The reviewer mentioned Texture Fields and GET3D in his comment and we agree that those papers should be properly contextualized. We added that contextualization for both papers in the Related Works (Section 2). Texture Fields shares similarities with our approach because they also define a color 3D space. However, the input and color spaces are different. Our method processes a mesh with UV-mapped textures, while Texture Fields demand a 3D shape and an object image, necessitating to render and unproject a depth map and thus being viewpoint dependent. Our approach uses the surface tubular neighborhood, defining color along normals of the entire surface, independently from viewpoints. Texture Fields also depends on additional networks to generate the latent code which conditions the Texture Field. Those networks are substantially more complex than the Texture Field itself, restricting its application to non-real-time scenarios. Specifically, the Texture Field is composed of 4 or 6 ResNet blocks, but the image encoder is a ResNet-18 for example. In contrast, our method adopts simpler architectures like small MLPs for efficient representation and rely on a GEMM-based customized algorithm to compute the normals. GET3D uses Texture Fields to represent the textures in its 3D model generation, thus the contextualization is analogous for this work.
>
> ### "Compare to a straightforward rendering of the more precise representation f, instead of transferring it".
>
> This is now better presented in Figure 10, which presents an perceptual evaluation of ground-truth meshes using rendered images of the meshes and our approach. Table 7 presents the MSE of between those images. All cases present error only in the second decimal digit.
>
> ### "The multi-scale sphere tracing algorithm is similar to the extension of marching cubes to multi-LOD presented in BACON."
>
> We made a better job at the Introduction to motivate the use of sphere tracing. The alternative to render neural SDFs is to extract a triangle mesh using marching cubes. However, due to the non-real-time nature of marching cubes, its application may be prohibitive in certain contexts involving neural SDFs. For instance, rendering dynamic SDFs requires mesh extraction at each time instant of the animation, posing a memory challenge if performed as preprocessing (extracted meshes use hundreds of MB in high resolutions while neural SDFs use a few KB). The performance is also a concern when surfaces are extracted while the animation is playing. Additionally, the smooth nature of the level sets of neural SDFs, which is a desired property in rendering, may be compromised by discretizing them into triangle meshes. Sphere tracing offers a solution to capture this smoothness. The algorithm is employed for rendering complex shapes, such as fractals and may use recent ray tracing hardware. Advancing neural SDF rendering through sphere tracing is an important topic rooted in classic Computer Graphics.
>
> ### "The GEMM implementation of normals computations should be compared to standard backpropagation in pytorch".
>
> We agree that this is of great importance and this version presents the comparison in Table 4. Our approach performs 2 times faster.

---

> ### Author Response · Authors · 2023-11-23
> **Part 2**
>
> ### "In the experiment section, when reporting training time vs NGLOD and IDF: which neural network architecture is used? Why call it “ours”? The whole method and contributions are about rendering neural fields with multiple levels of details, never about a new architecture or a faster training procedure."
>
> The reviewer is right, the previous version did not have enough details about the training. However, we have a specific training procedure now described in more detail in Section 3.3 (Nesting the Neighborhoods). We train h1, h2, f from fine to coarse and use the previously trained SDFs as ground truth for the next. That is, we consider f as ground truth for h2, which in turn is ground truth for h1. During the training of h2, we evaluate |h2 − f |∞, and choose the training step with the smallest norm. We do the same for h1. We note that the resulting norm |h2 − f |∞ is small during most of the training, a consequence of |h2 − f |^2 being a term in our loss function. Additionally, the above procedure provides a nested sequence that gives a robust rendering. Besides the resulting deltas being small (0.02 and 0.006 in Fig. 3), we note that decreasing them results in loss of silhouette (Fig. 3 (c)).
>
> ### At the end of the first paragraph of Sec 5.1 : “the attribute g is constant along the path since ||∇f|| = 1”: why does a gradient with constant norm implies a constant gradient?
>
> Sorry, we forgot to say that the path is the one with minimal distance to the surface, i.e. a straight line. We have fixed it.

---

### Official Review · Reviewer_Y2D7 · 2023-10-29

**Soundness:** 2 fair
**Presentation:** 3 good
**Contribution:** 3 good
**Rating:** 6
**Confidence:** 4

**Summary:**

The paper addresses the problem of real-time rendering for neural SDFs. It has 3 major contributions.
First, since neural SDFs are expensive to evaluate, the sphere tracing algorithm becomes too time-consuming. This work introduces multiscale sphere tracing where simpler MLPs are used to represent coarser SDFs which can be used for earlier iterations of sphere tracing.
Next, the paper introduces neural attribute mapping to allow normals and textures encoded in space to be mapped onto SDF or mesh surfaces. This techniques enable higher quality rendering.
Next, the paper improves normal computation for MLP-based neural SDFs with a GEMM-based algorithm.

**Strengths:**

- The paper introduces a novel acceleration method for sphere tracing of neural SDFs and a fast implementation for MLP gradients.
- The concepts and theories of multiscale sphere tracing and nested SDFs are explained clearly.
- The method does not rely on spatial data structures, thus is more suitable for representing dynamic neural surfaces.

**Weaknesses:**

- The paper does not make the setups of experiments sufficiently clear.
  - The method and experiments section of this paper explains the architecture of the proposed nested SDF representation, however, it is hard to see what are the inputs to the experiments and what are the losses without reading appendix (A.3).
- The comparison with Instant-NGP (Table 2) seems misleading.
  - If I understand the setting of the experiment correctly, the proposed method is trained with ground-truth SDF and color supervision, while the Instant-NGP is trained with multi-view image supervision. These are drastically different settings.
  - The images for Instant-NGP are generated with a renderer, thus COLMAP should not be required since it is used for camera pose estimation. Thus the speed comparison seems to be misleading.
- Missing comparisons with other representations.
  - Instead of comparing with Instant-NGP, the proposed nested SDF representation should be compared to a SDF representation with spatial data structure. For example, NanoVDB from the OpenVDB package.
- No performance comparison with the original mesh representation.
  - Ideally, the paper should show the advantages of the proposed representation over triangle meshes in rendering captured objects. However, the training data seem to come from mesh data.
  - Even though multiscale sphere tracing is 2x faster than sphere tracing with the original high resolution SDF, ~40 FPS still seems unusable in real applications compared with mesh-based representations.

**Questions:**

Most of my questions are described in the Weaknesses section. In addition, I have a few technical questions.
- Where does the speedup of the proposed GEMM normal computation come from compared to a autodiff framework? Is there improvements in algorithmic complexity, or it is mainly a reduction in overhead (e.g., better use of GPU resources)
- Is the GEMM normal computation differentiable to be used in an optimization framework?

---

> ### Author Response · Authors · 2023-11-23
>
> The reviewer has our most sincere thanks for helping improve our paper. We have a improved version after considering the valuable comments. We kindly ask the reviewer to read it. Next we discuss the review comments.
>
> ### "The paper does not make the setups of experiments sufficiently clear. The method and experiments section of this paper explains the architecture of the proposed nested SDF representation, however, it is hard to see what are the inputs to the experiments and what are the losses without reading appendix (A.3)."
>
> We agree that more detail about training was necessary. We now explain our training procedure in more detail in Section 3.3 (Nesting the Neighborhoods). We train h1, h2, f from fine to coarse and use the previously trained SDFs as ground truth for the next. That is, we consider f as ground truth for h2, which in turn is ground truth for h1. During the training of h2, we evaluate |h2 − f |∞, and choose the training step with the smallest norm. We do the same for h1. We note that the resulting norm |h2 − f |∞ is small during most of the training, a consequence of |h2 − f |^2 being a term in our loss function. Additionally, the above procedure provides a nested sequence that gives a robust rendering. Besides the resulting deltas being small (0.02 and 0.006 in Fig. 3), we note that decreasing them results in loss of silhouette (Fig. 3 (c)).
>
> ### "The comparison with Instant-NGP (Table 2) seems misleading."
>
> The set of advices regarding Instant-NGP in this review were very important. Trying to find a SOTA method to compare against our approach, we overlooked a few important details. We removed that comparison agreeing that the approaches are different in nature and that we should not have used COLMAP when running Instant-NGP in our case. Since our approach is the first one to compute textures in real-time for neural SDFs, the comparison in the updated paper is against the ground truth textured meshes, presented in Table 7 which shows the MSE between the generated images. The error of our approach is in the second decimal digit for all cases. Figure 10 shows the images used to calculate the MSE.
>
> ### "Ideally, the paper should show the advantages of the proposed representation over triangle meshes in rendering captured objects. However, the training data seem to come from mesh data. ... Even though multiscale sphere tracing is 2x faster than sphere tracing with the original high resolution SDF, ~40 FPS still seems unusable in real applications compared with mesh-based representations."
>
> The reviewer is right. We use triangle meshes as they provide surface and attribute data, which we need for our framework. However, we made a better job at the Introduction to motivate the use of sphere tracing. The alternative to render neural SDFs is to extract a triangle mesh using marching cubes. However, due to the non-real-time nature of marching cubes, its application may be prohibitive in certain cases involving neural SDFs. For instance, rendering dynamic SDFs requires mesh extraction at each time instant of the animation, posing a memory challenge if performed as preprocessing (extracted meshes use hundreds of MB in high resolutions while neural SDFs use a few KB). Performance is also a concern when surfaces are extracted while the animation is playing. Additionally, the smooth nature of the level sets of neural SDFs, which is a desired property in rendering, may be compromised by discretizing them into triangle meshes. Sphere tracing offers a solution to capture this smoothness. The algorithm is employed for rendering complex shapes, such as fractals and may use recent ray tracing hardware. Advancing neural SDF rendering through sphere tracing is an important topic rooted in classic Computer Graphics.
>
> ### "Where does the speedup of the proposed GEMM normal computation come from compared to a autodiff framework? Is there improvements in algorithmic complexity, or it is mainly a reduction in overhead (e.g., better use of GPU resources)"
>
> That is a core question and proper evaluation was lacking in the submitted version. In the updated version we compare our approach against autodiff in Table 4. Our approach performs 2 times faster. We do not think that this is a result of algorithmic complexity. We believe both approaches may be O(network inference). However, autodiff needs to perform a forward pass and to create and maintain a computational graph before being able to compute the gradient. Our approach just needs the input points and the network weights, calculating the normals in a modified forward pass. It needs to compute a forward pass for each one of the 3 normal coordinates, but those passes are independent and parallelized in Algorithm 2.
>
> ### Is the GEMM normal computation differentiable to be used in an optimization framework?
>
> That is a great question. We believe it is since we only need matrix multiplications and Hadamard products, and the activation function is smooth (periodic).

---

### Official Review · Reviewer_4bTK · 2023-10-30

**Soundness:** 3 good
**Presentation:** 3 good
**Contribution:** 3 good
**Rating:** 5
**Confidence:** 2

**Summary:**

The paper introduces a nested neighborhood model for real-time neural SDF rendering, emphasizing the efficiency of their normal computation method, based on General Matrix Multiply (GEMM) operations, which eliminates the need for auto-differentiation and computational graphs. The evaluation focuses primarily on Instant NGP and offers intuitive visualizations of the rendering results.

**Strengths:**

1. The concept of normal computation using GEMM operations without relying on auto-differentiation or computational graphs holds promise for real-time rendering applications.
2. The paper is well-written and includes mathematical analysis, although as a reviewer without a strong background in computer graphics, I cannot thoroughly evaluate the rigor of the mathematical aspects.
3. The evaluation against Instant NGP provides compelling evidence of efficiency, and the visualizations are visually impressive.

**Weaknesses:**

The major concern is the lack of reproducibility due to insufficient implementation details.

**Questions:**

1. The details provided may make it challenging to reproduce the results. Providing an implementation or a demo for readers to run would enhance the paper's credibility and make the results more accessible.
2. Table 4 suggests that a larger MLP leads to a more accurate zero-level set. It would be beneficial to discuss the theoretical limits or boundaries for achieving perfect results with different MLP sizes.

---

> ### Author Response · Authors · 2023-11-23
>
> We sincerelly thank the reviewer for the evaluation of our work. We updated the paper with several improvements guided by the insights provided by the review commitee and we politely ask the reviewer to consider reading it. Next we discuss the reviewer specific comments.
>
> ### "Providing an implementation or a demo for readers to run would enhance the paper's credibility and make the results more accessible."
>
> We agree with the reviewer. Our plan is to release the code before the publishing date. We believe that the algorithms defined in the paper will help popularize neural SDFs as a compelling surface representation in Computer Graphics.
>
> ### "Table 4 suggests that a larger MLP leads to a more accurate zero-level set. It would be beneficial to discuss the theoretical limits or boundaries for achieving perfect results with different MLP sizes."
>
> The reviewer is right, we provide now in Table 1 the Hausdorff distances between the ground-truth meshes and the surfaces of the neural SDFs and they suggest that larger MLPs could result in more accurate surfaces. However, our experience with SIREN networks is that they are capable of representing a huge frequency spectrum even with modest sizes. We speculate that the reason is because of the composition of periodic activation functions, which is capable of creating very complex sinusoids really faster. Thus, a very large MLP probably would result in a sinusoid too complex, which could be interpreted as noise. We suspect this is the reason why previous works have been reporting bad results using SIRENs. Those works tend to use networks too large.

---

### Official Review · Reviewer_Kd2D · 2023-11-01

**Soundness:** 2 fair
**Presentation:** 2 fair
**Contribution:** 2 fair
**Rating:** 5
**Confidence:** 3

**Summary:**

This paper tries to achieve real-time rendering of neural signed distance fields (SDF) with attribute mapping, like normal map and texture map. The core of its algorithm is multi-scale sphere tracking through nested fields with level-of-details. It claims to accelerate the computation of surface normal using an efficient GEMM-based implementation. The paper also extends the same set of tools for dynamic neural SDF.

**Strengths:**

- This paper considers lots of useful aspects of neural rendering, level-of-details, texture mapping, normal mapping, and multi-scale sphere tracing without spatial data structure. And this paper uses the concept of nested neighbors to combine these concepts together, making it interesting to read.
- The proposed algorithm is fast and capable of rendering high-resolution details, as demonstrated through several overfitting experiments.

**Weaknesses:**

- Some extra experiments can be added to provide more insight into the proposed method:
    - The author does not provide an evaluation of the speedup or the proposed GEMM-based implementation of surface normal computational.
    - In general, this algorithm uses more memory (coarse surfaces and finer surfaces) to trade speed. So, how does the speed compare with using extra spatial data structures, like bounding volume hierarchy for faster Ray surface intersection?
    - It is better to replace the MSE metric with PSNR when evaluating the image reconstruction quality, for example, in Table 8.
- The author needs further evidence to justify the concept of "nested SDF sequence" in this paper. There are two crucial question related to this concept
    - The author does not explicitly bound the sequence of SDF, neither through training regularization nor some explicit normalization. The concept of nested SDF sequence is only mentioned in deriving the algorithm, but in the experiments, it is not guaranteed to be nested. Does it mean that for most of the popular neural SDF with level-of-detail design, without any explicit control of the bound of nested SDF, the proposed algorithm will always converge effectively?
    - The tightness of the bound and how the tightness affects the algorithm is not analyzed and demonstrated.

**Questions:**

- I don't understand why the GEMM-based implementation will accelerate the normal computational. I thought auto-diff will also use GEMM to compute derivatives. Can you explain it more? Also, measuring the speed of normal computations will make this point more solid.
- In most of your experiments, you only use two nested SDF. What if you use more? Like, 3 to 5, even to 10 neural SDF?

---

> ### Author Response · Authors · 2023-11-23
>
> Please accept our deepest thanks for helping us to improve our paper with the very insightful comments. We politely ask the reviewer to read the updated version which includes substantial improvements. We will discuss about the changes related with this specific review.
>
> ### "The author does not provide an evaluation of the speedup or the proposed GEMM-based implementation of surface normal computational."
>
> We agree that this evaluation is important and was missing in the previous version. We compare our approach with the standard autograd normal computation in the updated version (Table 4). Our approach is consistently 2 times faster.
>
> ### How does the speed compare with using extra spatial data structures, like bounding volume hierarchy for faster Ray surface intersection?
>
> We compare against NGLOD, which uses uses a sparse voxel octree (SVO) to represent the neural SDF and to render its zero-level set using a sparse sphere tracing algorithm. Specifically, the vertices of
> the voxels store features. In Figure 5 we show that one implication of that approach is discretization artifacts caused by non-continuous gradients. Our approach is not only continuous when using SIRENs, but it is smooth, since it calcules the attributes directly from the smooth neural SDF (smoothness is guaranteed by SIREN's periodic activation).
>
> ### "The authors does not explicitly bound the sequence of SDF, neither through training regularization nor some explicit normalization. The concept of nested SDF sequence is only mentioned in deriving the algorithm, but in the experiments, it is not guaranteed to be nested. Does it mean that for most of the popular neural SDF with level-of-detail design, without any explicit control of the bound of nested SDF, the proposed algorithm will always converge effectively?"
>
> We agree that the paper should explain that better. The updated version has several improvements in that direction. We now explain our training procedure in more detail in Section 3.3 (Nesting the Neighborhoods). We train h1, h2, f from fine to coarse and use the previously trained SDFs as ground truth for the next. That is, we consider f as ground truth for h2, which in turn is ground truth for h1. During the training of h2, we evaluate |h2 − f |∞, and choose the training step with the smallest norm. We do the same for h1. We note that the resulting norm |h2 − f |∞ is small during most of the training, a consequence of |h2 − f |^2 being a term in our loss function. Additionally, the above procedure provides a nested sequence that gives a robust rendering. Besides the resulting deltas being small (0.02 and 0.006 in Fig. 3), we note that decreasing them results in loss of silhouette (Fig. 3 (c)).
>
> We have a theorem (Prop. 1 in Section 3.3) which provides tight upper bounds for the deltas. Section 3.4 has an informal proof that the sphere tracing converges when using the deltas computed using Prop. 1.
>
> ### "In most of your experiments, you only use two nested SDF. What if you use more? Like, 3 to 5, even to 10 neural SDF?"
>
> That question is legitimate and now instead of a sequence with m neural SDFs, we our definitions suppose a sequence of 3 SDFs {h1, h2, f}, corresponding to the cases in our experiments. In practice, we may choose how to use functions h2 and f to adapt to a specific performance budget. For example, we may choose not to iterate using h2, but directly map the normals of the surface in f to the surface in h1. This setup relies on sphere tracing iterations using the coarse SDF, decreasing the cost per iteration. Another possible case is to use h2 for a better surface approximation, and map normals of the surface of f as before. We can also use h2 and f for sphere tracing and do not perform normal mapping. This setup has the best silhouette.
>
> ### "I don't understand why the GEMM-based implementation will accelerate the normal computational. I thought auto-diff will also use GEMM to compute derivatives. Can you explain it more? Also, measuring the speed of normal computations will make this point more solid."
>
> The reviewer is right to assume that auto-diff uses GEMM to compute the derivates, however it also needs to perform a forward pass before that to compute the computational graph, and also has the additional overhead of creating and maintaining that graph. Our approach does not need any additional passes nor graphs, just the input points and the network weights. In the updated paper we compare our approach with the standard autograd normal computation (Table 4). Our approach is consistently 2 times faster.

---

### Meta-Review · Area_Chair_BPct · 2023-12-07

**Metareview:**

The paper presents a novel nested neighborhood model designed for real-time rendering of neural signed distance fields (SDF) with attribute mapping, including normal maps and texture maps. The algorithm's essence lies in multi-scale sphere tracking, employing simplified Multilayer Perceptrons (MLPs) to characterize coarser SDFs. These coarser representations are utilized in the initial stages of sphere tracing. Additionally, the paper enhances normal computation for MLP-based neural SDFs through General Matrix Multiply (GEMM) operations.

The paper comprehensively explores various crucial facets of neural rendering, including level-of-details, texture mapping, normal mapping, and multi-scale sphere tracing without spatial data structures. Leveraging the concept of nested neighbors, the proposed algorithm seamlessly integrates these aspects. It exhibits impressive speed and the ability to render high-resolution details. The concept of normal computation using GEMM operations without auto-differentiation or computational graphs has promise for real-time rendering applications. In addition, the method's independence from spatial data structures makes it particularly well-suited for representing dynamic neural surfaces.

The paper unnecessarily employs complex formalism. The neural attribute mapping problem it tackles seems either previously explored in other papers or should be contrasted with a more straightforward rendering approach. The experiments are not convincing, and comparisons with traditional pipelines are absent. Furthermore, the comparison with Instant NGP appears irrelevant, given the substantial differences in problem settings. Instant NGP focuses on learning scene representation solely from images, while the proposed method has access to a supervision signal for both 3D geometry and RGB texture. The paper falls short in adequately describing both implementation and experimental details, hindering the reproducibility and fair comparison of the proposed method. Despite some addressed concerns in the rebuttal, the reviews generally maintain their original reservations.

**Justification For Why Not Higher Score:**

Experiments are not convincing. In general, the paper uses overly complex mathematics for very simple ideas that have already been exploited. Most reviewers hold reservations.

**Justification For Why Not Lower Score:**

N/A

---

### Decision · Program_Chairs · 2024-01-16

Reject